# Adsorption and Desorption studies of *Delonix regia* pods and leaves: Removal and recovery of Ni(II) and Cu(II) ions from aqueous solution

**Bolanle M Babalola[1], Adegoke O Babalola[2], Cecilia O Akintayo[1], Olayide S. Lawal[1], Sunday F Abimbade[1], Ekemena O. Oseghe[3], Olushola S Ayanda[1, *]**

[1]Department of Industrial Chemistry, Federal University Oye-Ekiti, Ekiti State, Nigeria
[2]Ekiti State Ministry of Works and Infrastructure Ado-Ekiti, Ekiti State, Nigeria
[3]Nanotechnology and Water Sustainability Research Unit, College of Science, Engineering and Technology, University of South Africa-Science Campus, Florida 1710, South Africa

[*]Corresponding author: osayanda@gmail.com, bolanle.babalola@fuoye.edu.ng

## Abstract

In this study, the adsorption of Ni(II) and Cu(II) ions from aqueous solutions by powdered pods and leaves of *Delonix regia* was investigated by batch adsorption techniques. The effects of operating conditions such as pH, contact time, adsorbent dosage, metal ions concentration and the presence of sodium ions interfering on the sorption process were investigated. The results obtained showed that the equilibrium sorption was attained within 30 min of interaction, an increase in the initial concentration of the adsorbate, pH and adsorbent dosage led to increase in the amount of Ni(II) and Cu(II) ions adsorbed. The adsorption process followed the pseudo-second-order kinetic model for all the metal ions sorption. The equilibrium data fitted well with both the Langmuir and Freundlich Isotherms, the monolayer adsorption capacity ($Q^0$ mg/g) of the pods and leaves of *Delonix regia* for Ni(II) ions was 5.88 mg/g and 5.77 mg/g, respectively, and 9.12 mg/g and 9.01 mg/g, respectively for Cu(II) ions. The efficiency of the powdered pods and leaves of *Delonix regia* for Ni(II) and Cu(II) ions removal was > 80%, except for the sorption of Ni(II) ions onto the leaves. The desorption study revealed that the percentage of metal ions recovered from the pods were higher than the leaves at various concentrations of nitric acid. This study has proven that *Delonix regia* biomass, an agro-waste could be used for removing Ni(II) and Cu(II) ions from aqueous solution.

*Keywords:* *Delonix regia*, low-cost adsorbent, Ni(II) ions, Cu(II) ions, equilibrium, kinetics

## 1. Introduction

The persistent nature, non-biodegradability, toxicity and ability to bio-accumulate in the environment have made heavy metals priority pollutants (Hamza et al., 2013). Various health effects are caused by anthropogenic pollutants in water; which are majorly heavy metals such as mercury, nickel, lead, cadmium copper, zinc and cobalt (Hamza et al., 2013; Singh et al., 2011). Heavy metals gain entrance into water resources by industrial activities such as electroplating, smelting, production of glass, textile, paper and ceramics, mining, textiles, storage batteries, petroleum, metal finishing, pulp and paper (Dean et al., 1972; Ksakas et al., 2018; Kumar et al., 2019). The damage caused by copper to the marine life include damage of gills, liver, nervous system, kidneys and changing the sexual life of fishes (Flemming and Trevors, 1989; Ho et al., 2002; Van Genderen et al., 2005). Although, copper is known to play a vital role in the metabolism in animal; its excessive intake can result in serious health problems (Paulino et al., 2006). The permissible limit of copper in wastewater and portable water is 0.5 mg/L and 2.5 mg/L, respectively (Zhou et al., 2018; Kumar et al., 2019). Reactive free oxygen species which damage lipids, proteins and DNA are released when copper is present in the blood system (Brewer, 2010). Wilson's and Alzheimer's diseases, mental illness, Indian childhood cirrhosis and schizophrenia are also reported to be caused by excess copper in the blood (Brewer, 2007; Faller, 2009; Hurean and Faller, 2009). Nickel has detrimental effects on human health, resulting in allergic dermatitis, immunologic urticarial; immediate and delayed hypersensitivity (Festus et al., 2013). All nickel compounds, except for the metallic nickel, have been classified as human carcinogens by the International Agency for Research on Cancer (IARC) (IARC, 1990) and the U.S. Department of Health and Human Services (DHHS, 1994). Due to its toxicity in minute quantity, 0.05 mg/L was set for the permissible limit of nickel in wastewater (Zhou et al., 2018).

Conventional methods for the removal of metal ions includes: chemical precipitation, oxidation/reduction, ion exchange, electrochemical processes, membrane separation, Fenton process, ozonisation, electrocoagulation, photochemical degradation and evaporation (Okoya et al., 2014). These techniques require high operational costs and yield minimal removal efficiencies, they have been reported to be expensive and inadequate. Therefore, there is the need to investigate alternative techniques that are cheaper, efficient and easy to handle. One of such techniques is biosorption, that is, the use of low-cost adsorbent like agricultural materials of no economic value and industrial by-products (Jeme, 1968; Inoue and Munemori, 1979) for the removal of heavy metal ions from polluted water.

Almond shells tree bark treated with formaldehye and sulphuric acid (Guibal et al., 1993; Raji
et al., 1997), bone char, tea leaves, wood charcoal (Ajmal et al. 2003), and coconut shells have
been used to produce activated carbon to remove heavy metal ions from wastewater.  Rice hulls
(Ajmal et al., 2003), rice bran (Montanher et al., 2005) and pine bark (Nath et al., 1997) have
also been used in the raw and treated form to remove heavy metal ions. The removal of Ni(II)
ions from aqueous solution using sugarcane bagasse, an agricultural waste biomass, has been
investigated by Garg et al. (2008). The dosage for maximum removal of Ni(II) ions from an
aqueous solution of 50 mg/L were reported to be at 1500 mg/L adsorbent dosage and at pH 7.5.
Moodley et al. (2011) investigated the adsorption capacity of pine sawdust by treating
wastewater containing Ni(II), and other metal ions (Co(II) and Fe(III) ions). The adsorption
and desorption of Ni(II) Ions from aqueous solution by a lignocellulose/montmorillonite
nanocomposite was reported by Zhang and Wang (2015). Their report indicated that the
maximum adsorption capacity of Ni(II) ions reached 94.86 mg/g at an initial Ni(II) ions
concentration of 0.0032 mol/L, a solution pH of 6.8, temperature of 70°C, and contact time of
40 min. Kahraman et al. (2008) examined the use of cotton stalk and apricot seeds as alternative
adsorbents for the removal of Pb and Cu. The removal of Pb and Cu by these agricultural wastes
was reported to reducing their toxic effects on *P. aeruginosa*. The sorption capacity of Cu(II),
kinetics and isotherms of different low-cost residual agricultural materials including peanut
shells, nut shells, plum seeds, eucalyptus bark, olive pips, peach stones, and pine sawdust was
studied by Hansen et al. (2010). Moreover, Abdel-Tawwab et al. (2017) used rice straw,
sugarcane bagasse, and maize stalks for the removal of Pb, Cd, Cu, and Zn from aqueous
solution. All the biosorbents were reported to be effective and cheap for the removal of the
metal ions from polluted water, with rice straw showing a higher adsorption efficiency than the
others. The application of treated pumpkin husk as an excellent adsorbent for removing Cu(II)
and Ni(II) ions has been reported by Samuel et al. (2016). The adsorption of Cu(II) and Ni(II)
ions was found to be suitable at pH 5.
*Delonix regia* also known as flame of forest, is a semi-deciduous tree, native to Madagascar. It
is popularly grown in Africa and Hong Kong as a shade tree and for ornamental purpose. The
tree has pods that can be as long as 60 cm in length and 5 cm wide, with a distinct bright green
fern-like compound leaves and attractive red peacock flowers. Researchers have reported the
usefulness of the green leaves and flowers of *Delonix regia* in medicine i.e. *Delonix regia* have
a broad spectrum of pharmacological activities in various ailments (Modi et al., 2016).
However, it sheds its leaves and flowers in dry areas and seasons, the trees are less attractive
after the leaves and flowers are shed, with their pods remaining on the branches until they are
dropped by wind, these make *Delonix regia* an agro-waste with limited valuable use.
There are limited studies on the use of *Delonix regia* for the removal of organic and inorganic
contaminants from aqueous solution, except for Ponnusami et al. (2009), Onwuka et al. (2016)
and Babalola et al. (2019) who have reported the viability of *Delonix regia* for the removal of
methylene blue dye, crude oil spill and Pb(II) ions, respectively. Therefore, the objective of
this research is to investigate the capacity of the pods and leaves of *Delonix regia* in the removal
of Ni(II) and Cu(II) ions from aqueous solutions. The desorption of bound metals from spent
*Delonix regia* pods and leaves using various concentrations of nitric acid was also considered.
**2. Materials and methods**
2.1 *Delonix regia* sample
Leaves and pods of *Delonix regia* collected from Ekiti State University, Ado-Ekiti, Nigeria
were used as adsorbent for the sorption study. The materials were washed with deionised water,
sun-dried and milled. After milling, the adsorbents were sieved through a 250μm mesh nylon
sieve and kept in air tight containers until required for use.
2.2 Chemicals and reagents
Diammonium nickel hexahydrate and copper chloride dihydrate salts supplied by Merck,
Germany were dissolved in high purity milli-Q water to prepare 1000 mg/L stock solutions of
Ni(II) and Cu(II) ions, respectively. Working standard solutions were prepared from the stock
solutions and pH adjustment was done with 0.1 M $HNO_3$ and 0.1 M NaOH when necessary.
The effect of solution ionic strength on sorption was studied using different concentrations of
sodium nitrate salt and desorption of bound metal from spent biomass was achieved with
different concentrations of $HNO_3$.
2.3 Characterization
The elemental composition of the pods and leaves of *Delonix regia* was achieved by energy
dispersive spectroscopy (EDS). The morphological study was by the scanning electron
microscope (Nova Nano SEM 230) and the transmission electron microscope (FEI Tecnai $G^2$
20). X-ray diffractometer (Siemens D8 Advance Bruker XRD) was used for the phase
characterization.


2.4 Adsorption

Adsorption procedure by Meena et al. (2008) was slightly modified and used in this work. Parameters such as the influence of pH, contact time, initial adsorbate concentration, adsorbent dosage and solution's ionic strength were investigated.

For the influence of pH, a 0.5 g of the biomass was weighed into designated tubes containing 20 mL of 100 mg/L metal ions solution of various pH in the range 1 to 8. The suspensions were shaken on an end-over-end shaker for 300 min at ambient temperature ($21\pm2^{o}C$). The influence of contact time was conducted by varying the contact time from 5 - 300 min at optimised pH. The concentration of Ni(II) and Cu(II) ions was varied from 1 - 1000 mg/L for the influence of initial metal ions concentration at optimised pH and contact time. Finally, the influence of adsorbent dosage and solution's ionic strength was achieved using 100 mg/L metal ions concentration, the parameters were varied from 0.25 - 1.0 g and 0.001 - 0.5 M $NaNO_3$, respectively at optimised pH and contact time. Each of the experiment was carried out in triplicate and the results were the average values.

At the expiration of each of the experiment, aliquot was taken, centrifuged (twice at 4500 rpm for 15 min and 10300 rpm for 10 min) and diluted with 0.1 M $HNO_3$ before the residual metal content was analysed by Thermo elemental inductively coupled plasma – mass spectroscopy (ICP-MS; X Series II).

2.5 Analytical procedure

Multi-elements standard solution was used to prepare different concentrations of external calibration standard solutions (1 – 100 µg/L) used for the analysis. After the instrument was switched to the operate mode, the pressure of the nebulizer was checked, and the instrument's sensitivity and stability were checked by running a tuning solution of 1 µg/L multi element standard solution for 10 min. It was ensured that the $^{115}In$ counts was less than 20 K counts per second, precision less than 2%. The backgrounds (5BKg and 220BKg) were less than 1 count per second, the oxides cerium oxide ($CeO^+$) less than 2.0% and the doubly charged ions ($Ba^{2+}$) less than 6.0%. Therefore, the calibration standard solution and the blank were analysed to obtain a calibration curve where the samples to be analysed will properly fit. The analysis of the samples proceeded after the calibration curve for each of the metal ions was good with $R^2$ = 0.9999.

The amount of Ni(II) and Cu(II) ions sorbed on the pods and leaves of *Delonix regia* ($Q_e$) was calculated using Equation 1.

$$Q_e = (C_o - C_e)V/m \qquad\qquad 1$$
where $C_o$ is the initial metal concentration (mg/L), $C_e$ is the final metal ion concentration in the
solution, ICP-MS reading (mg/L), V is the volume of the metal solution used in litre (L) and
m is the mass of the biosorbent (g).
2.6 Kinetics and Equilibrium modelling
Further examination of the contact time experiment was carried out by modelling the data with
the pseudo-second-order kinetic models to determine the mechanism of the sorption process.
The pseudo-second-order kinetic model can be expressed in Equation 2 where $k_2$ (g/mg/min)
represent the rate constant for the pseudo-second-order kinetics; $Q_e$ and $Q_t$ (both in mg/g) being
the amount of adsorbate taken up by the adsorbent at equilibrium and at time t, respectively.
All parameters in Equation 2 were derived by plotting $t/Q_t$ against t.
$$^t/_{Q_t} = {}^1/_{k_2} + {}^t/_{Q_e} \qquad\qquad 2$$
The data obtained on the effect of initial metal ions concentration was also modelled using the
Langmuir and Freundlich isotherms. Monolayer surface coverage, availability of equal number
of adsorption sites on the adsorbent and no interaction between adsorbed species were
assumptions of the Langmuir model (Liu and Wang, 2014). The Langmuir Isotherm is
represented in Equation 3
$$\frac{C_e}{Q_e} = \frac{1}{Q^0 b} + \frac{C_e}{Q^0} \qquad\qquad 3$$
where $Q_e$ (mg/g), is the amount of solute adsorbed per unit mass of adsorbent; $C_e$ (mg/L) is the
equilibrium concentration of solute in the bulk solution, $Q^0$ (mg/g) represent the monolayer
adsorption capacity of the adsorbent and $b$ (L/mg) represents the constant related to the energy
of adsorption. The Langmuir equation could be further expressed using a dimensionless
constant separation factor $K_R$ shown in Equation 4. $K_R$ is a relationship containing all the
essential features of the Langmuir isotherm.
$$K_R = \frac{1}{1 + K_a C_i} \qquad\qquad 4$$
where $C_i$ is the initial concentration of metal ions in solution (mg/L) and $K_a$ stands for the
Langmuir constant (L/mg). This dimensionless separation factor is interpreted to imply that
the isotherm is favourable if $0 < K_R < 1$.
The Freundlich isotherm states that the ratio of the amount of solute adsorbed onto a given
mass of the adsorbent to the concentration of the solute in the solution is not constant at
different concentration. The Freundlich isotherm is used to describe adsorption onto
heterogeneous surfaces (Equation 5).
$$logQ_e = logK_F + \frac{1}{n}logC_e \qquad\qquad 5$$
where 1/n is the constant related to the adsorption efficiency, $K_F$ is the adsorption capacity, $Q_e$
is the quantity of adsorbate adsorbed per unit weight of the adsorbent and $C_e$ is the final
concentration of metal in the solution. The Freundlich constant 1/n is a factor giving an
indication of how favourable the adsorption of the adsorbate onto the sorbent is, $0 < 1/n < 1$
imply favourable adsorption. It is also related to the heterogeneity of the adsorbent surface.

**3. Results and discussion**
3.1 SEM-EDS, TEM and XRD analyses
The SEM (at x10 000 magnification) and TEM micrographs of the pods and leaves of *Delonix*
*regia* (Fig. 1) indicated the spongy nature of *Delonix regia* with porous structures, being
potentially beneficial for the uptake of the Ni(II) and Cu(II) ions from aqueous solution. The
EDS of *Delonix regia* showed that the surfaces of the leaves was composed of 66.79% C,
32.97% O and traces of Ca and K, while, the pods was composed of 57.61% C, 41.15% O and
trace of K.

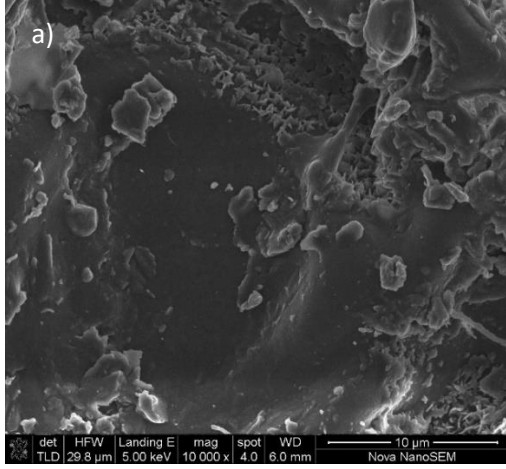 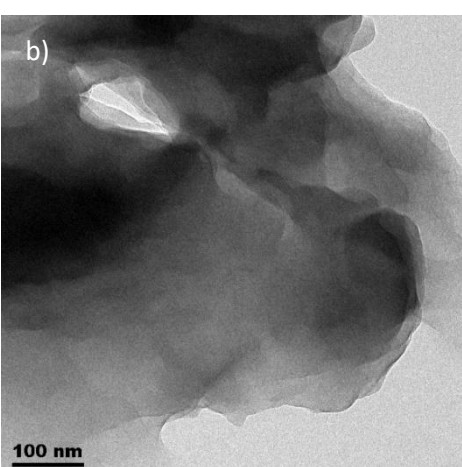

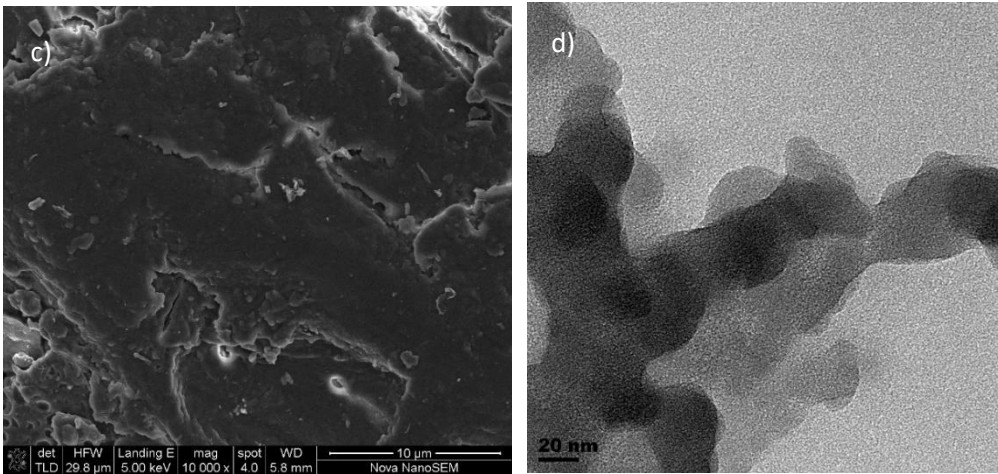


**Figure 1.** SEM (at x10 000 magnification) and TEM micrographs of *Delonix regia* leaves (a
& b) and pods (c & d)

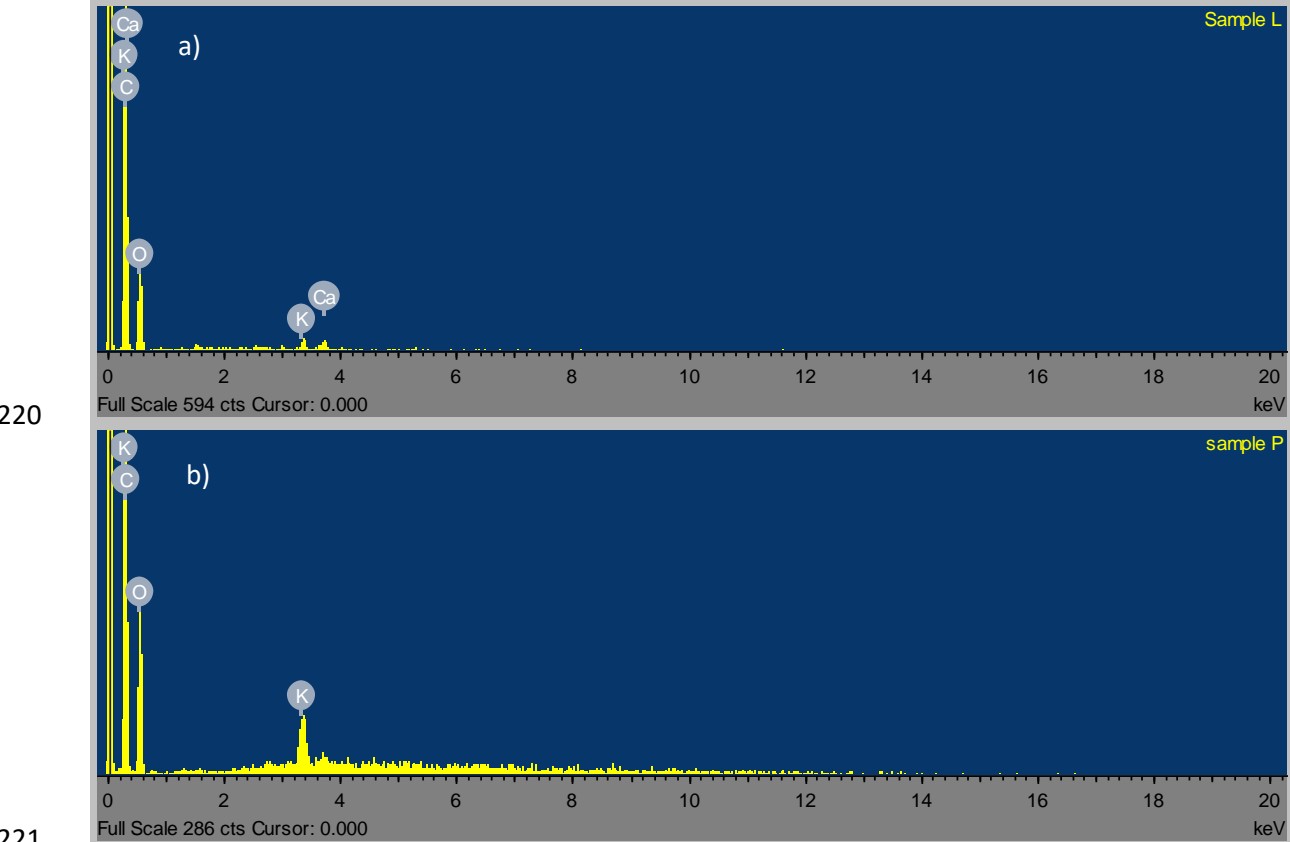



**Figure 2.** EDS of leaves (a) and pods (b) of *Delonix regia*
According to the ICSD Nos. 03-0289, 20-0231 and 26-1077, native cellulose ($C_6H_{12}O_6)_x$) and
whewellite, syn ($C_2CaO_4 \cdot H_2O/CaC_2O_4 \cdot H_2O$) were identified as major peaks in *Delonix regia*
(Fig. 3). The pods were mainly native cellulose, whereas, the leaves were rich in whewellite,

syn. The hemicellulosic moieties in the pods and leaves might provide sites for the binding of Ni(II) and Cu(II) ions.

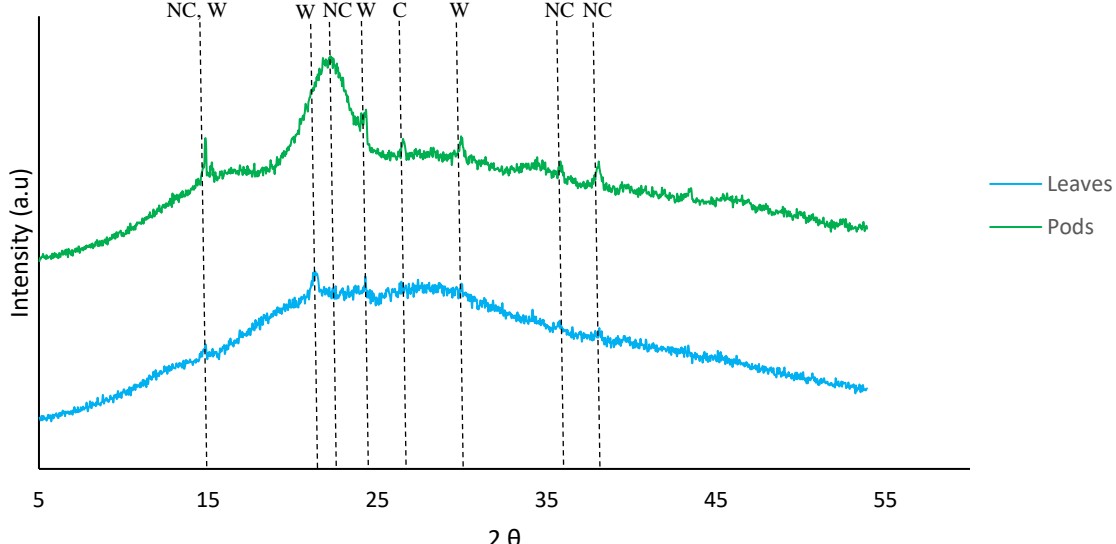

**Figure 3.** X-ray diffractogram of the leaves and pods of *Delonix regia*

(NC – native cellulose ($(C_6H_{12}O_6)_x$); W – whewellite, syn ($C_2CaO_4 \cdot H_2O/CaC_2O_4 \cdot H_2O$); C – carbon)

3.2 Adsorption studies

3.2.1 pH

The result of the study of pH on the adsorption of Ni(II) and Cu(II) ions onto *Delonix regia* is shown in Fig. 4. The figure reveals that there was no evident nickel uptake by both adsorbents at low pH, but a very noticeable uptake was achieved as pH increased. pH 5 at which maximum sorption was recorded was used in subsequent experiments. Saeed et al. (2005b) reported an optimum sorption pH of 6 for Ni(II) on crop milling waste while pH 5 was reported for its sorption onto natural neem sawdust and almond husk (Hasar, 2003; Rao et al., 2007), thus being in line with our findings.

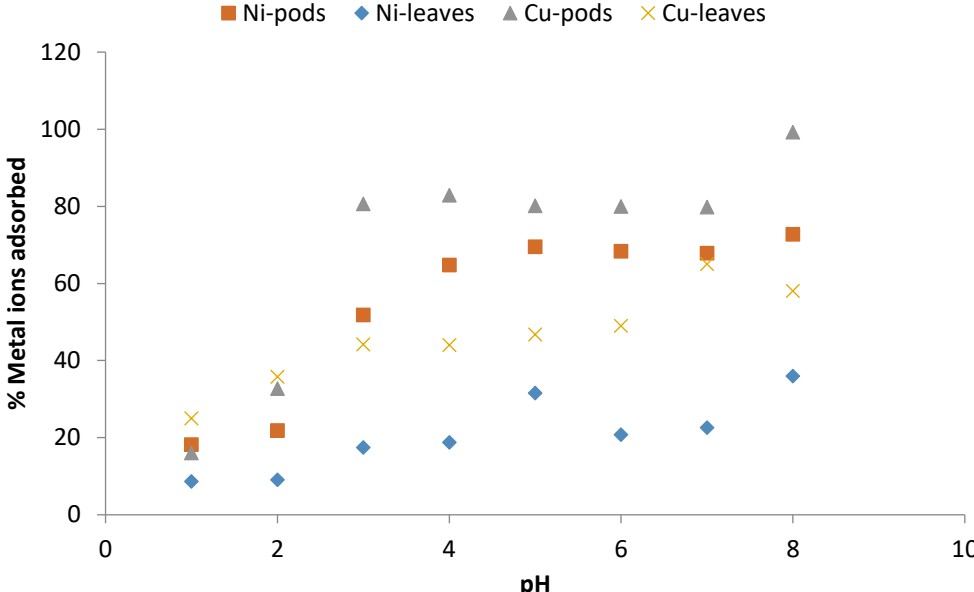

243
**Figure 4.** Effect of pH on the adsorption of Ni(II) and Cu(II) by *Delonix regia* pods and leaves.
*Experimental conditions:* pH 1-8; contact time: 300 min; adsorbent dosage: 0.5; metal ions
concentration: 100 mg/L

A small increase in the uptake value was achieved between pH 1 to 3, but maximum uptake of

46.7% was recorded at pH 5 for Cu(II) ions sorption onto leaves and maximum uptake of 82.9%

at pH 4 onto the pods which were further used for adsorption of Cu(II) onto the leaves and

pods, respectively. The increase recorded at pH 6 and 7 were not considered in choosing the

optimum pH at which Cu(II) ions sorption onto the leaves occurred because visible precipitate

which might be due to the formation of $Cu(OH)_2$ or other soluble complexes had been formed

in the experimental set up at these pH (Larous and Meniai, 2012). Some authors reported pH 5

and 5.8 as the optimum for the uptake of Cu(II) ions by some modified natural wastes (Shukla

and Pai, 2005; Witek-Krowiak et al., 2011; Anantha and Kota, 2016). This might be as a result

of the increased overall negative charge particularly between pH 4 and 6 which would

subsequently lead to increase the sorption of the positive metal ions. The increase of pH

decreases the concentration of hydrogen ions and therefore the competition between metal ions

and hydrogen ions for active sites on the adsorbent is reduced (Kadirvelu et al., 2000; Kadirvelu

et al., 2001; Sánchez-Polo and Rivera-Utrilla, 2002; Kadirvelu et al., 2003; Meena et al.,

2005a).

    3.2.2 Contact time

The investigation of the effect of agitation time performed at different time intervals of 5 min
up to 300 min showed that the two adsorbents used in this work had rapid uptake of the metal
ions within short period of interaction (Fig. 5a).

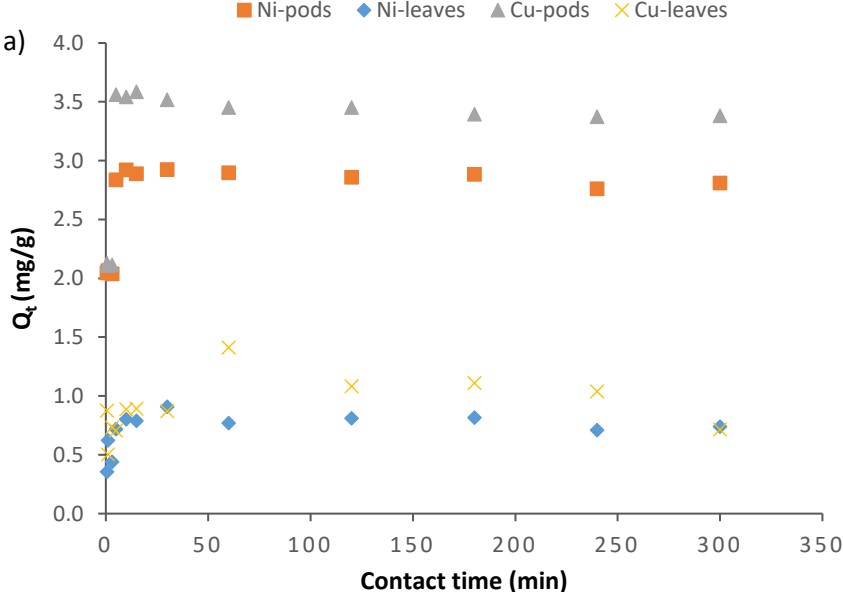


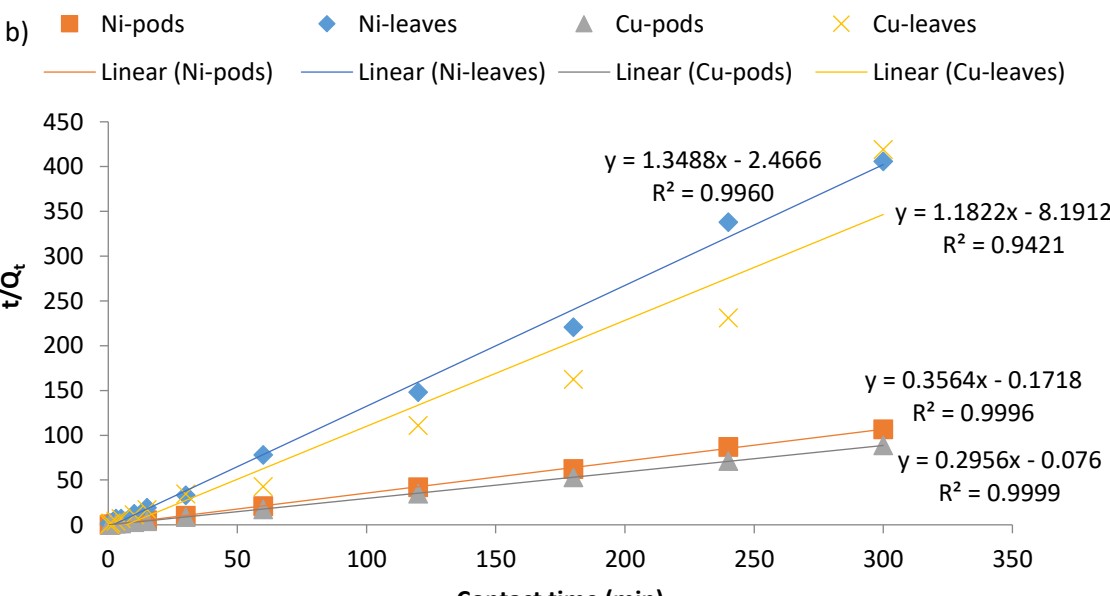

**Figure 5.** Effect of contact time (a) and pseudo-second-order kinetics (b) of the adsorption of
Ni(II) and Cu(II) ions by *Delonix regia* pods and leaves. *Experimental conditions:* pH 4 and 5;
contact time: 5-300 min; adsorbent dosage: 0.5 g; metal ions concentration: 100 mg/L

A slight desorption and adsorption was observed between 5 to 10 min of interaction of the
adsorbent with the metal ions solution. Within 30 min of interaction, maximum uptake has

been achieved on both adsorbents irrespective of the metal ions considered, though the uptake recorded on the pods were higher than those on the leaves. The native cellulose, a major component of the pods might be responsible for the enhanced uptake on the pods. Metal uptake by the adsorbents remained fairly constant after 30 min of interaction until the end of the experiment. Moreover, the maximum uptake at 30 min could be considered to be faster when compared with other biosorbents reported in the literature. Olufemi and Eniodunmo (2017) reported 30 min and 120 min as the optimum time for the removal of Ni(II) ions from aqueous solutions using coconut shell and banana peels, respectively. Ksakas et al. (2018) reported 60 min and 180 min as the optimum contact time for the sorption of Cu(II) ions unto some types of natural clay. Alatabe (2018) reported 120 min as the optimum sorption time for Cu(II) ions onto activated carbon from *Cane papyrus*. However, 30 min was the optimum contact time reported by Kumar et al. (2019) on the removal of Cu(II) ions by groundnut seed cake powder, sesame seed cake powder and coconut cake powder.

The kinetic modelling of data showed that the process was well fitted to the pseudo-second-order kinetic model. This is supported by Hansen et al. (2010) who reported that the adsorption of Cu(II) ions onto various agriculture waste materials fitted the pseudo-second-order kinetic model, the plots and kinetic parameters of this model are shown in Fig. 5b and Table 1, respectively. The pseudo-second-order kinetic model indicates that the mechanism involved in the sorption is governed by ion exchange or sharing of electrons (Babalola et al., 2019).

**Table 1**. Parameters obtained from the pseudo-second-order kinetic model

| Metal ions | $Q_e$ (mg/g) | $k_2$ (g.mg$^{-1}$min$^{-1}$) | $R^2$ |
|---|---|---|---|
| *Delonix regia* pods | | | |
| Ni | 2.8 | -2.36 | 0.9996 |
| Cu | 3.4 | -3.24 | 0.9999 |
| *Delonix regia* leaves | | | |
| Ni | 0.7 | -0.25 | 0.9960 |
| Cu | 0.8 | -0.09 | 0.9421 |

In a similar study by Babalola et al. (2019), the calculated $Q_e$ value from the pseudo-second-order kinetic model of the adsorption of Pb(II) ions onto *Delonix regia* pods and leaves are 4.12 mg/g and 2.7g mg/g, respectively. Bansal et al. (2009) reported 2.81 mg/g for the sorption of Ni(II) ions onto rice husks.

3.2.3 Initial adsorbate concentration and isotherm experiment
The experiment conducted to study the effect of the initial concentration of adsorbate on the
uptake of Ni(II) and Cu(II) ions by the pods and leaves of *Delonix regia* showed that the uptake
of both metal ions on the two types of adsorbent increased with increasing adsorbate
concentration (Fig. 6a). When adsorbate concentration is increased, there is increased driving
force of the metal ions to the binding sites of the adsorbents and thus increased uptake is
recorded. Though, the uptake per unit gram of the adsorbent increased, the opposite which is a
decrease was observed in the percentage of metal removed from the solution as the adsorbate
concentration increased. This is because as the ratio of the metal ions to the adsorbent increases,
the exchangeable sites in the adsorbent structure are saturated, resulting in a decrease in the
percentage removed (Man et al., 2012). The data obtained in Figs. 5a and 6a also suggested
that the pods are better than the leaves and the adsorption is more favourable onto the pods at
all concentrations used in this study.

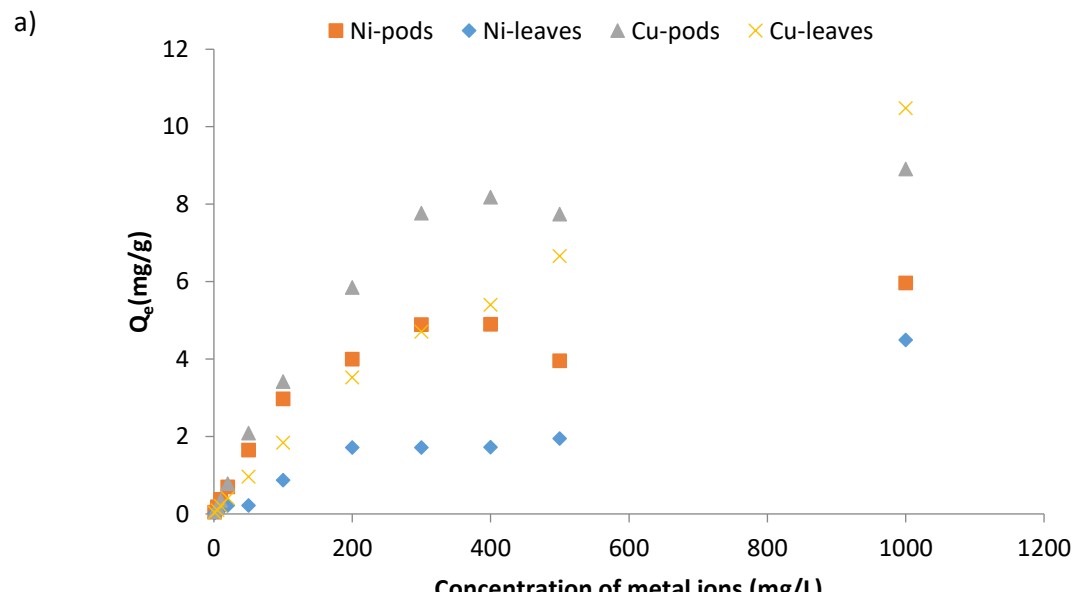


b)

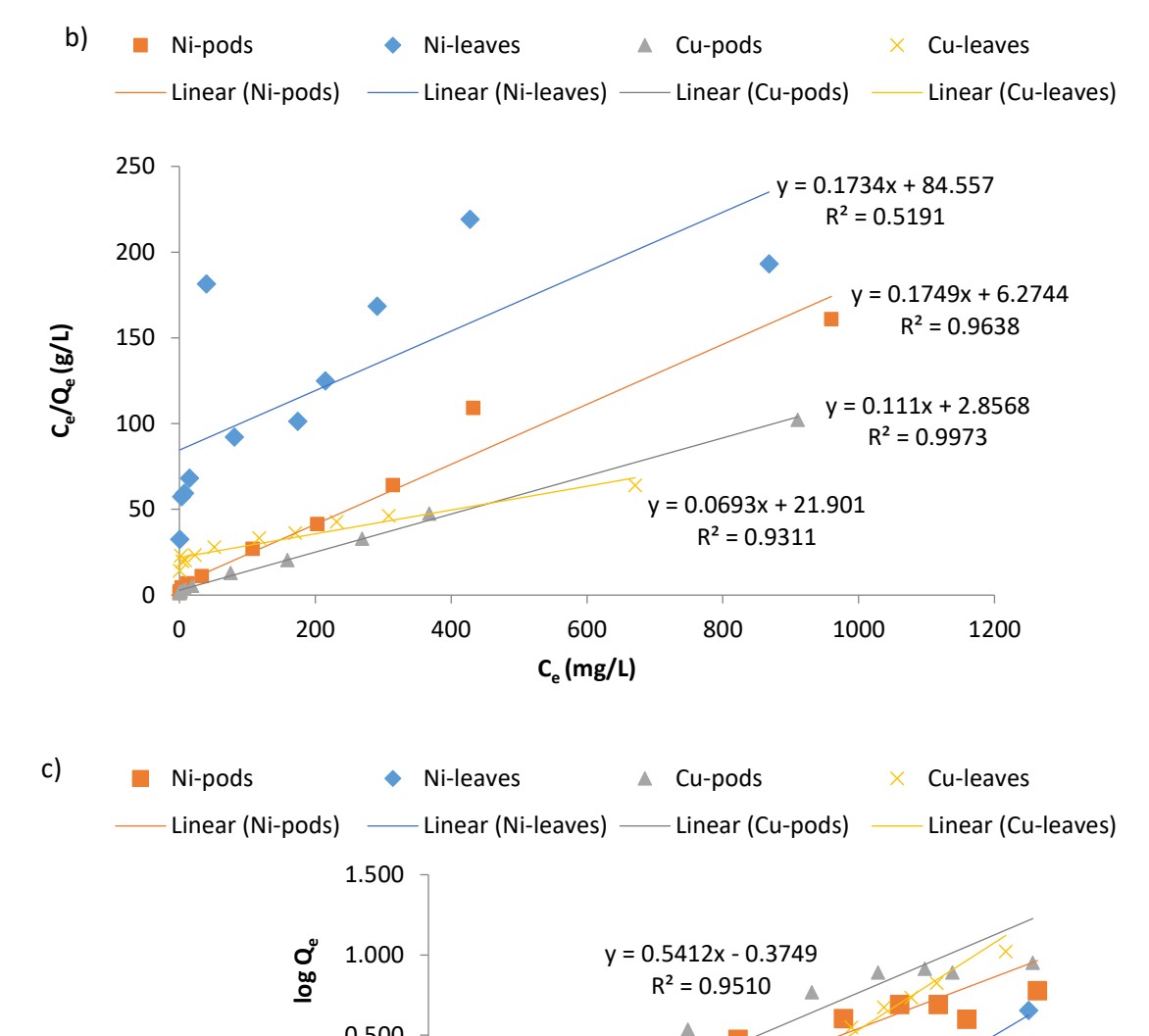


c)

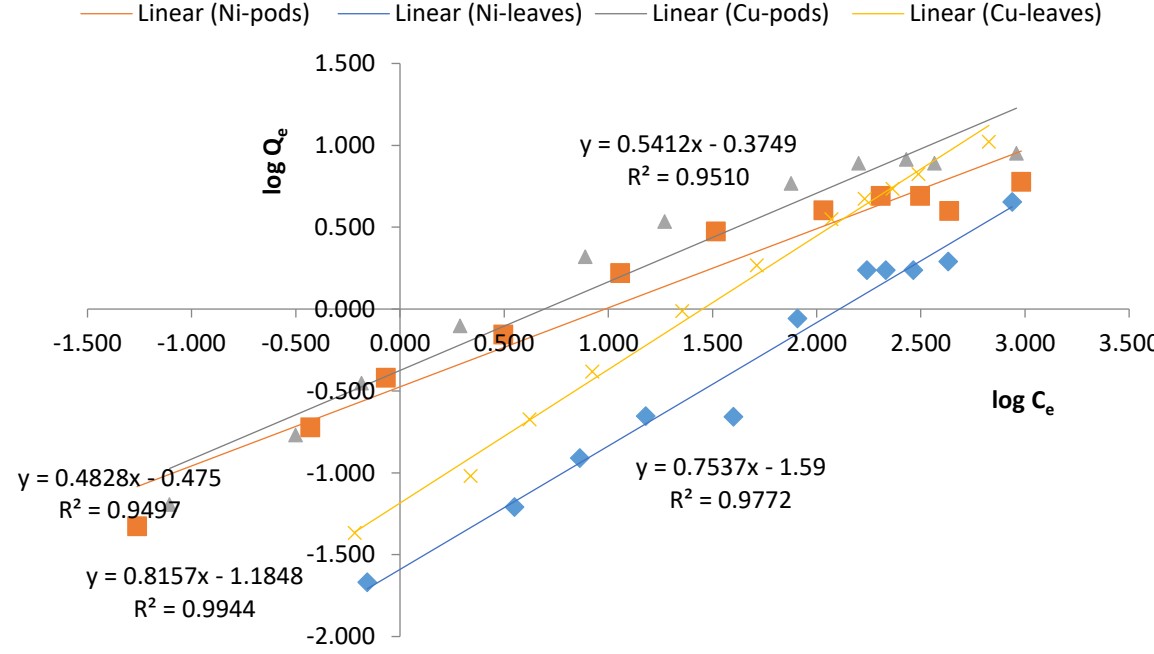


**Figure 6.** Effect of initial metal ions concentration (a), plots of Langmuir (b) and Freundlich
isotherm (c) isotherms for the adsorption of Ni(II) and Cu(II) ions by *Delonix regia* pods and
leaves. *Experimental conditions:* pH 4 and 5; contact time: 30 min; adsorbent dosage: 0.5 g;
metal ions concentration: 1 - 1000 mg/L

324

The relatively high concentration of metal ions compared to the adsorbent available binding surfaces as concentration is increased could be responsible for the decrease observed in percentage removal (Meena et al., 2005b; Hamza et al., 2013). The data obtained from this experiment was modelled using the Langmuir and Freundlich isotherm models.

The results presented in Fig. 6b for the Langmuir Isotherm model showed that for the pods the Langmuir isotherm is favourable for its adsorption of Ni(II) and Cu(II) ions. The values obtained for the adsorption of the metal ions onto the leaves showed that adsorption of Cu(II) ions unto the leaves is more favourable than that of Ni(II) ions. The fit to the plot of the Langmuir isotherm suggests the possible monolayer coverage of the metal ions on the adsorbent surface except for the sorption of Ni(II) ions to the leaves where fluctuations were observed. The high correlation coefficient ($R^2$) obtained for the isotherm when sorption was carried out using the pods of *Delonix regia* is also an indication of its applicability in the sorption reaction. The correlation coefficient obtained using *Delonix regia* leaves as adsorbent were not as high as those obtained as the correlation for the pods, implying that Langmuir Isotherm is not the best model to explain the sorption of Ni(II) and Cu(II) onto the leaves. The parameters obtained for the Langmuir isotherm modelling are shown in Table 2.

**Table 2.** Isotherm models adsorption parameters for the adsorption of Ni(II) and Cu(II) onto *Delonix regia*

| Metal ions | Langmuir Isotherm | | | | Freundlich Isotherm | | |
|---|---|---|---|---|---|---|---|
| | $Q^0$ (mg/g) | b (L/mg) | $R^2$ | $K_R$ | $K_F$ (mg/g) | 1/n | $R^2$ |
| **Pods** | | | | | | | |
| Ni | 5.88 | 0.02 | 0.9638 | 0.32 | 0.66 | 0.34 | 0.9497 |
| Cu | 9.12 | 0.03 | 0.9973 | 0.24 | 0.86 | 0.39 | 0.9510 |
| **Leaves** | | | | | | | |
| Ni | 5.77 | 0.0021 | 0.5191 | 0.83 | 0.03 | 0.75 | 0.9772 |
| Cu | 9.01 | 0.0389 | 0.9311 | 0.21 | 0.07 | 0.81 | 0.9944 |

The plot of the Freundlich isotherm is presented in Fig. 6c. Using the Freundlich isotherm model on the data obtained from the concentration study, it was further discovered from the values of the adsorption parameters shown in Table 2 that the adsorption of Ni(II) and Cu(II) ions onto *Delonix regia* leaves is more favourable than the adsorption of these metal ions onto the pods. Values obtained for 1/n are 0.75 and 0.81 for the adsorption of Ni(II) and Cu(II) ions, respectively for the leaves while the values for their respective sorption onto the pod are 0.34

and 0.39, respectively. The values of the Freundlich constant 1/n which are lower for the pods
and a higher adsorption capacity, $K_F$ for the pods are further confirmation that the pods adsorb
better than the leaves.
The values obtained for the Freundlich isotherm correlation coefficient ($R^2$) during the
adsorption of Ni(II) and Cu(II) ions onto the leaves are higher than those obtained for the
adsorption onto the pods. Thus, the Freundlich isotherm is a better isotherm to describe the
adsorption of Ni(II) and Cu(II) ions onto *Delonix regia* leaves, while, the Langmuir isotherm
is better fitted to describe adsorption onto the pods.
Kumar et al. (2019) reported that the sorption of Cu(II) ions onto groundnut seed cake, sesame
seed cake and coconut cake powders fit perfectly to the Langmuir isotherm with $Q^0$ ranging
from 3.608 to 3.703 mg/g. Likewise, Saeed et al. (2005a) reported the biosorption of Cu(II)
ions to perfectly fit the Langmuir adsorption isotherm model. Olufemi and Eniolaodunmo
(2018) reported a $Q^0$ of 1.47 mg/g and 2.57 mg/g for the sorption of Ni(II) ions onto banana
peels and coconut shells, respectively, with Langmuir constant *b* (L/mg) as 0.018 and 0.016,
respectively. Malkoc and Nuhoglu (2005) recorded a $Q^0$ of 15.26 mg/g and *b* value of 0.088
L/mg for the adsorption of Ni(II) ions onto tea factory waste. Thus, the isotherm experiment
has shown that the Langmuir constant related to the sorption energy obtained in this study were
very close to banana peels, coconut shells and tea waste, except for the adsorption of Ni(II)
ions onto the leaves which was lower. Moreover, the $Q^0$ of the pods and leaves of *Delonix regia*
in this work were higher than that of banana peels and coconut shells, but lower than what was
obtained for the tea factory waste.
In a similar study by Babalola et al. (2019), the Langmuir isotherm provided the best fit for the
adsorption of Pb(II) ions onto the pods of *Delonix regia*, whereas, the Freundlich isotherm also
gave a better fit for the leaves. This implies that the sorption of these metal ions onto the pods
assumes a monolayer adsorption onto a homogeneous surface with a finite number of identical
sites, whereas, sorption onto the leaves assumes that the metal ions adsorb onto the
heterogeneous surface of the adsorbent (Ayanda et al., 2013).
3.2.4 Adsorbent dosage and solution's ionic strength
Fig. 7a revealed the results of the effect of changing the adsorbent doses on the adsorption of
Ni(II) and Cu(II) ions by *Delonix regia* biomass. It was observed that the percentage adsorption
of Ni(II) ions increased from 45.9% to 80.4% and from 21.6% to 33.7% onto the pods and
leaves, respectively, whereas, the percentage adsorption of Cu(II) ions onto both adsorbent

increased from 72.6% to 89.2% and from 69.6% to 80.8% onto the pods and leaves, respectively. The observed increase in the percentage Ni(II) and Cu(II) ions adsorption might be due to the availability of more binding sites as adsorbent doses are increased.

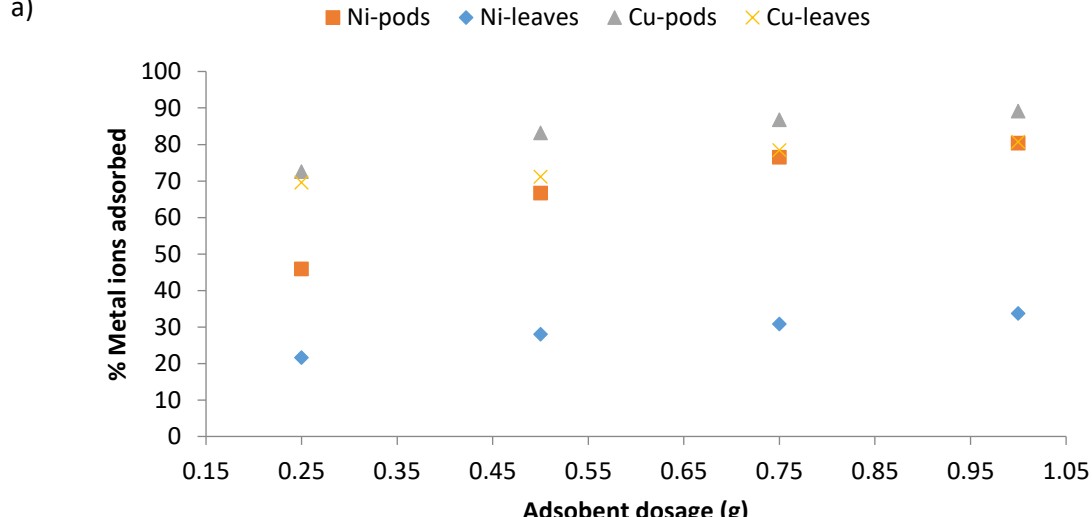

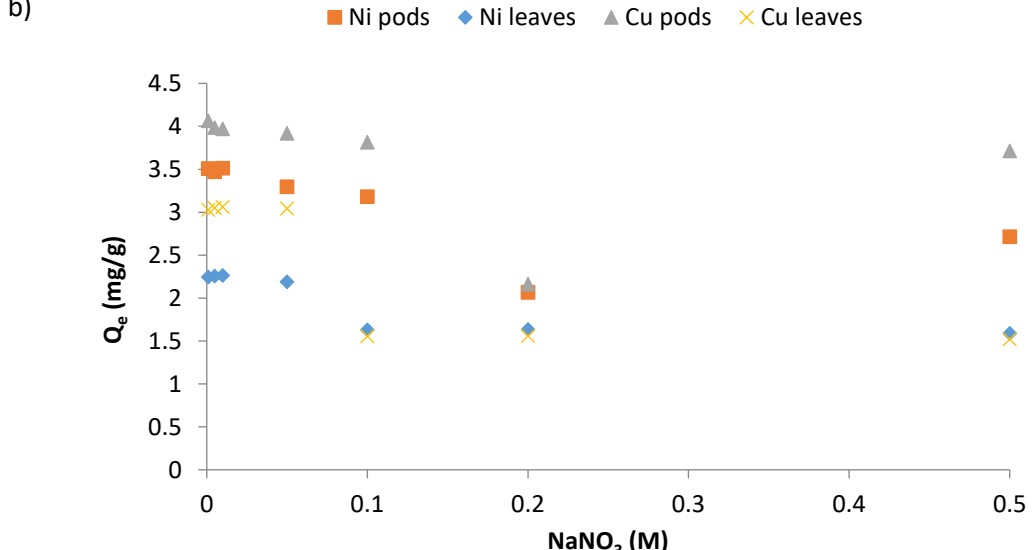

**Figure 7.** Effect of adsorbent dosage (a) and ionic strength (b) on the adsorption of Ni(II) and Cu(II) ions by *Delonix regia* pods and leaves. *Experimental conditions:* pH 4 and 5; contact time: 30min; metal ions concentration: 100 mg/L

The result of the effect of sodium ion on the adsorption of Ni(II) and Cu(II) ions by *Delonix regia* biomass is shown in Fig. 7b. The figure shows a reduction in the uptake of both metal

ions by each of the biomasses. It could be explained that there was competition for the available
binding sites on the adsorbents by the positive sodium ions present in the adsorption medium
(Alegbe et al., 2019). Thus, as the concentration of sodium ion is increased metal uptake by the
adsorbent was observed to be lower.
A comparison of the adsorption efficiency of different agricultural waste for Ni(II) and Cu(II)
ions is presented in Table 3.

**Table 3**. Comparison of the amount of Ni(II) and Cu(II) ions removed by some agro-waste
with the pods and leaves of *Delonix regia*

| Agricultural waste | Metal ions | Adsorbent dosage | Efficiency | References |
|---|---|---|---|---|
| Tea waste | 100 mg/L Ni(II) | 10 g/L | 86% | Malkoc and Nuhoglu, 2005 |
| Saw dust | 10 mg/L Ni(II) | 5 g/100 ml | 75% | Shukla et al., 2005 |
| Sugarcane bagasse | 50mg/L Ni(II) | 1500 mg/L | > 80% | Garg et al., 2008 |
| Banana peel | 100 mg/L Ni(II) | 4.5 g/50 mL | 78% | Olufemi and Eniolaodunmo 2018 |
| Coconut shell | 100 mg/L Ni(II) | 4.5 g/50 mL | 75% | |
| Papaya wood | 10 mg/L Cu(II) | 5 g/L | 95% | Saeed et al., 2005a |
| Rice bran | Cu(II) | 200 mg/20 mL | >80% | Montanher et al., 2005 |
| Groundnut seed cake | | | | |
| Sesame seed cake | 10 mg/L Cu(II) | 0.75-1 g/50 mL | 99.7% | Kumar et al., 2019 |
| Coconut cake | | | | |
| Spruce Sawdust | 10 mg/L Cu(II) | 1 g/100 mL | >85% | Kovacova et al., 2019. |
| *Delonix regia* pods | 100 mg/L Ni(II) | | >80% | |
| *Delonix regia* leaves | 100 mg/L Ni(II) | 0.5 g/20 mL | >30% | This study |
| *Delonix regia* pods | 100 mg/L Cu(II) | | >85% | |
| *Delonix regia* leaves | 100 mg/L Cu(II) | | >80% | |


The results obtained from this study have shown that *Delonix regia* biomass compete
favourably with other agricultural waste used in previous studies and hence, could be useful in
removing Ni(II) and Cu(II) ions from aqueous solution.

3.3 Desorption Study
The regeneration of adsorbents after adsorption is of outmost importance, metal ions adsorbed
should be easily desorbed under suitable conditions and the adsorbents should be repeatedly
use to reduce the cost of the material. Thus, the recovery experiment was carried out to
investigate the possibility of recovering the adsorbed metal ions from the biomasses. The result
obtained from the recovery study is shown in Fig. 8.

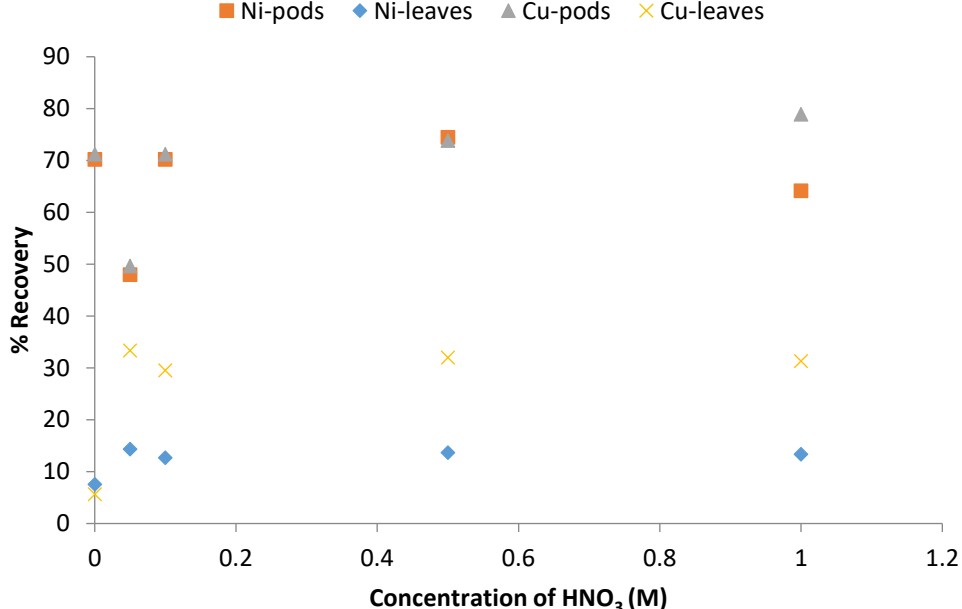

**Figure 8.** Recovery study of the adsorbed Ni(II) and Cu(II) ions from the pods and leaves of
*Delonix regia*

From Fig. 8, the values obtained for the percentage recovery of Ni(II) and Cu(II) ions from the
pods of *Delonix regia* showed that different percentages of metal ions were recovered at
different concentrations of the desorbing medium. The results also revealed that relatively low
concentration of nitric acid (0.1 M) could be used to recover more than 50% of the metal ions
from the pods. The figure also revealed that the percentage of metal ions recovered from the
pods was higher than what was recovered from the leaves at various concentration of nitric
acid. Approx. 74.4% Ni(II) ions and 78.9% Cu(II) ions were recovered from *Delonix regia*
pods with 1.0 M nitric acid concentration, . whereas, 14.3% Ni(II) ions and 33.3% Cu(II) ions
were recovered from the *Delonix regia* leaves with the same concentration of nitric acid (1.0
M). The recovery experiment showed that metal ions recovery increased with increasing
concentration of nitric acid. Moreover, *Delonix regia* biomass, most especially the pods could
be repeatedly used in the removal of heavy metal without losing its adsorption properties.
**4. Conclusion**
The study concerns the use of powdered *Delonix regia* pods and leaves for the removal of
Ni(II) and Cu(II) ions from aqueous solutions. The results obtained indicated that the pH of the
solution, contact time, initial metal ion concentrations, adsorbent dosage and ionic strength
affect the uptake of the metal ions by the biosorbent. It was observed that the Freundlich
isotherm is a better isotherm to describe the adsorption of Ni(II) and Cu(II) ions onto *Delonix*
*regia* leaves, while, Langmuir isotherm is better fitted to describe the adsorption onto the pods.
The pseudo-second-order kinetic model agrees with the sorption of Ni(II) and Cu(II) ions onto
*Delonix regia* pods and leaves. The desorption study also showed that metal ions could be
desorbed from spent *Delonix regia* and could be repeatedly used in the removal of heavy metal
without losing its adsorption properties. The pods performed better than the leaves in terms of
the amount of metal ions removed and regeneration of the spent adsorbent. In conclusion, the
powdered pods and leaves of *Delonix regia* could be used as an eco-friendly, cheap and
effective adsorbents for the removal of Ni(II) ions, Cu(II) ions and other environmental
contaminants from aqueous solution.

**Data availability**
The data generated and/or analysed during the current research are available with the authors
upon reasonable request.

**Author contributions**
BMB was the investigator and contributed to writing the paper. AOB and EOO were involved
in the characterization of the adsorbent. COA, OSL and SFA were involved in the adsorption
studies. OSA was involved in the modelling of the adsorption data and contributed to the
writing of the paper.

**Conflict of Interest**
On behalf of the authors, the corresponding author states that there is no conflict of interest.

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
