# Peer review of "Adsorption and Desorption studies of *Delonix regia* pods and leaves: Removal and recovery of Ni(II) and Cu(II) ions from aqueous solution"

_Drinking Water Engineering and Science, 2019_

## Referee Comment (RC1) · Anonymous Referee #1 · 19 Mar 2020

Adsorption and desorption studies of Delonix regia pods and leaves: removal and recovery of Ni(II) and Cu(II) ions from aqueous solution. The removal of Ni and Cu was studied by adsorption of specific agro-waste. It is an interesting study, but the paper should be modified and the written quality of the paper should be improved, and apart from the given suggestions of the reviewer, be checked by a native speaker. General comments: - A clear objective (and knowledge gap) at the end of the introduction is still missing or weak. How does this relate to previous research in the area? The "therefore" in line 99 is not well underpinned/ - Language, including tenses, is of poor

quality. - Redundant information should be deleted. - The discussion of the results is weak. How do the results link to other research? How are the results (including isotherm constants and kinetic constants) relate to other adsorbents? - The order of the presented results should be reconsidered. Better to start with the kinetic tests and equilibrium tests and then relate that to change in pH and structure of the pods and leaves. Specific comments: - Line 15: include stating that it were "batch tests" - Line 17-18: it is not relevant in the abstract if data fit a model. More important is how well the metals adsorb in relation to other adsorbents. - Line 20: "concentrations". However, it is not clear from the data if nitric acid is effective. - Line 21-22: delete sentence, since a good economic study is not made - Line 38: delete "of heavy metals" - Line 39: delete "and cause serious pollution" - Line 40-42; delete sentence - Line 43: supply = resources - Line 44: add "production" ( of glass, textiles, paper etc.) - Line 45: delete "apart from.. ecosystem" - Line 46: insert "human" (gills, liver….) - Line 48"" during = in the; of animal = in animals - Line 52: add "also" (reported) - Line 53: delete "Like most heavy metals"; its = a - Line 54, add "," after "health" - Line 55 add "," after "compounds" and after "nickel" - -line 58-61: too simple, many processes are effective and thus not "inadequate". So refer better to other research. Byt the way: reverse osmosis is a membrane process… - Line 64, adee "," after "value" - Line 66: "shells, " - Line 67: "shells" - Line 68: "treat" = "remove"; indicate effectiveness of the shell as well - Line 69: what was the "treatment" of the materials like? - Line 71: insert "," after "biomass"; "optimum conditions" = "dosage" - Line 73: delete "adsorbent dosage of"; insert "at" before pH: stirring speed is not relevant - Line 75-78: delete sentence - Line 83: "seeds" - Line 84, delete "the treatment …. To"; "reduce" = "reducing" - Line 88-90: delete sentence - Line 96-98: delete sentence - Line 99-: give knowledge gap, based on the previous research and explain reason to study selected material better. - Line 106-108: delete sentence (not relevant in the context of the research) - Lin 111: give concentrations of stock solutions - Line 118-121: give pHs to be studied: give concentrations that are studied - Line 129: "use" = "used" - Line 130-131: "is" = "was" (three times) - Line 132: "are" = "were"; do not indicate only average but also the

ranges - Line 129-137: explain procedure of the batch tests in detail (number of jars, concentrations, dosages); further explain the modelling part in the M&M section too. - Line 141: "structures, being potentially beneficial for the uptake" - Line 143-144: "is" = "was" (two times) - Line 144-145: what do these data indicate? Give discussion with literature - Line 156: how can we find the value in the Figure? - Line 158-159: why the authors have this opinion? Can they support this with literature? - Line 166: explain in M&M what were dosages and how it was known that equilibrium was reached, (may be start with kinetic tests.). - Line 167: "reveals"; do not use "significant" without statistical analyses.. - Line 168: "adsorbents"; delete "very" (subjective); "increased". - Line 169: delete sentence - Line 170: "is" = "was'; delete "as the optimum pH and used" - Line 173: add ", thus being in line with our findings" and explain why. - Line 175 (and further): do not connect data points with lines in figures (no meaning) - Line 180: delete "for Cu(II) ions"; "pH 4 was used . . . pods while pH 5 was" = "which were further". - Line 181: "leaves and pods, respectively" - Line 182-184: discuss with literature what happened. - Line 184: delete "Thus. . . studies"; "literatures" = "authors" - Line 187: delete "Generally. . .. Media". - Line 188: give reference - Line 189-191: delet ""when. . ... particles" - Line 192: add "the positive metal ions"; "decreases" - Line 194: delete "This will. . . adsorbent" - Line 201: "for" = "of" - Line 203: see comment of line 175 - Line 209: "interacting" = "interaction with" - Line 212: "adsorbents" - Line 213: "to" = "until" - Line 217: explain better how the work of Hansen supports the data - Line 221: "is a. . . fact" = "indicates"; give reference for this statement - Line 223-226: should be in Materials and methods section - Line 223: "is well" = "can be" - Line 225: delete "any" - Line 228: explain how the results relate to research on other adsorbents, so discuss with literature - Line 235: "decreased"; this statement is normal for isotherms, so please give references - Line 236-237: rephrase sentence - Line 246: "higher number" = "high concentration"; give also a reference for this statement - Line 248-249: delete sentence (normal behaviour) - Line 250:263: should be explained in Materials and methods section - Line 264- 267: the fact that the pods are better than the leaves is a result of the data and not of the isotherms. So better first discuss the

data and then the isotherms - Line 269: "The fit to the plot of the Langmuir isotherm suggests the possible monolayer..." - Line 271: delete "slight" - Line 272-274: delete sentence - Line 277: explain why the Langmuir isotherm does not fit well to the leaves data. Discuss this with literature. - Line 280: discuss the obtained data with data form literature on other adsorbents - Line 283: "The Freundlich isotherm is used to describe adsorptions onto.." - Line 288-293: part of Materials and methods secion - Line 295-299: see observation in Line 264-267. - Line 302-304: explain why this is with literature and how this relates to other research. How can it be that the pods fit to both Langmuir and Freundlich and the leaves only to Freundlich? Is can be a question of equilibrium concentration. - Line 305 and further: changing adsorbent dose does not give extra information over changing concentrations. These are both methods do determine the isotherms - Line 321: "shows" - Line 322: "biomasses" - Line 322-325: give references for this statement - Line 331: see comment on Line 175 - Line 335: "concentrations" - Line 336: "Was" = "were" - Line 337-339: what can be the reason? Discuss with literature - Line 340:342: discuss with literature - Line 344-346: the effect of higher concentrations are not clear, so this cannot be concluded - Line 348: "application" = "use" - Line 352-354: Explain the reason. - Line 355-356: see line 344-346 - Line 357-359: this cannot be concluded because a thorough economic analysis is not made.

---

## Referee Comment (RC2) · Anonymous Referee #2 · 25 Mar 2020

Interactive comment on: Title: Adsorption and Desorption studies of Delonix regiapods and leaves: Removal and recovery of Ni(II) and Cu(II) ions from aqueous solution Author(s): Bolanle M. Babalola et al.

General comments The manuscript presents adsorption studies of Cu(II) and Ni (II) in pods and leaves of a plant. The adsorption parameter effects of contact time, kinetics, isotherms and ionic strength are presented. Besides, experiments of desorption were performed. The approach is interesting; however, the document has to be improved in the way of presenting the information/results, comparing with other literature (newer

one) and avoiding presenting the same data in both graph and tables (for example kinetic data), the graphs quality has to be improve. It is recommended to compare with other adsorbents in terms of adsorption capacity (mg/g) and not percentage and including the initial concentrations. The aim or need of the study is not well stated, the application is only for wastewater? The conclusions both, pods and leaves, are promising ones to remove the metals or only one of the plant parts? How should the material be applied, in batch or filters? No studied about economics was performed, so better not to mentions economic aspects.

Specific comments Abstract: Line 13-14 mention that conditions such as pH, contact time, metal ions concentration and the presence of sodium ions were studied, however in the abstract only the results of kinetics is presented, the results of the others conditions should be included.

Line18-19: present the results of the isotherm data, at least the capacity in terms of mgMetal/gadsorbent and state is it is good or not in relation with others adsorbents in literature.

Introduction:

Include newer references as the newest one is from 2016. Some information about reports, monitoring or regulations of the metals in drinking and wastewater is needed, so, there will be an idea of which concentrations are of concern and should be used in the study to evaluate the adsorbents, that would justify why 100mgM/L was used in the study. Line 66-98 the removal capacity effectiveness of many adsorbents is reported, however almost no data of the capacity and initial concentrations of the metals is included, include the mg/g if that information is reported. That information gives an idea of why to go for natural adsorbents. Line 99 explain why is this research needed and why to choose this adsorbent, why to check the leaves and the pods?.

Materials and methods.

Lines 111-112 were the leaves and pods dried? Line 117 what were the concentrations used? Line 120-121 include the sodium and nitic acid concentrations tested. Line 119 instead of referencing to paper include the procedures in detail including for example: type of water used, adsorbent dosage, metal concentrations, rpm, volume of solutions, time, etc. Besides it would be better to include subitems for each experiment starting from: kinetics, pH effect, isotherms, ionic strength effect and desorption experiment. Each one explained in detail. Were the experiments performed at least in duplicate? Line 134 Equation 1 is not relevant, it is better to include after each subitem mentioned in the previous comment the corresponding equations for kinetics and isotherms. Better here than in the results and discussion section. Line 136 a specific subsection is needed for the analitycal procedure including equipment brand and model and the detection limit for both metals.

Results and discussion Lines 156-158 The diffractogram first shows that both materials are amorphous, that could be related to high surface area and potential adsorption. Besides, include in Figure 3 the diffractogram of cellulose so, that it can be directed compare, specially thinking that in natural products not only cellulose is present, what about lignin for example? DRX is not enough to characterize a natural products, other techniques like IR, MNR, etc is needed. What is found in literature for natural adsorbents of this type? Which moiety are responsible for the adsorption? Line 157 mention amino group, but in figure 2 no nitrogen was detected. Line 164-165 start with the kinetic study. Line 165 it is important to mention that in general pods performed better than leaves for both metals in the whole pH range, why? Line 176: include adsorbent dose. Line 183: what can be the precipitate? What literature says? Line 200 the equipment and time evaluated goes in the materials and methods section. Line 201 it is evident that pods are better than leaves, make a comment. Line 207 include adsorbent dose. Line 205 -228 the information in Fig. 5b and Table 1 is basically the same, let the Table 1 in the document. Line 210-213 then adsorption was mainly in 30 min, how is that compare to the literature? Faster, slower? Line 218 how Hansen et al. (2010) supports your results on kinetics? Line 221-222 needs a reference and an

explanation on how the ion exchange or sharing of electrons explains the adsorption of the metals in the adsorbent. Line 223: put the equation in materials an methods. Line 228: compares the Qe and k values obtained for the pods and the leaves among them and the literature. Line 230 this is confusing is that the same experiment for the isotherms? Then, the title item should refers to the isotherms experiments. Line 244: what is the adsorbent dose? Line 241 regarding with Fig. 6b and Fig 6c those are the linearized isotherms and the same information is on Table 2. Recommend to plot Qe vs Ce instead of those figures. Line 254 as mentioned before, including those equations in the materials and methods, include the actual isotherms equations and the linearized ones. Line 272 repeated word: correlation Line compare data of Table 2 with others adsorbents in literature. Line 296-300 the data suggest that the pods are better than the leaves, it can be seen from the graphs in Figs. 5a and 6a, from the data. Besides, 1/n and Kf are lower and higher respectively for the pods for both metals confirming that the pods perform better and adsorption is more favorable on them at the concentrations studied. Line 300-304 higher R2 only confirms which data fit better to the model not which one is a better or a favorable adsorption. Line 305 the adsorbent dose experiment is to determine the isotherm. Line 316 better to use normal scale, not logarithmic so the effect is easier to see. Line 331 was that experiment as all made by duplicate? The results with 0.05M HNO3? Are confusing, how is that the Ni and Cu pods % is substantially lower and with ultra pure water and higher concentration is practically the same in both cases? How to explain that? Line 349-351 what implications has those results in the application of those materials as adsorbents. Line 352-352 what can be concluded from the isotherm data? Are those good or not for the removal of the metals, are them better than others? Line 354/355 what to conclude from the kinetic, is it fast or lower than others? Line 355/356 see comment on line 331, confusing data. Line 357-359 no economical study was done, so can't conclude that. Besides only two metals were evaluated and using ultrapure water, so can't conclude is application in natural conditions and besides, can't conclude about others environmental contaminants.

---

## Author Comment (AC1) · 2 Apr 2020

Response to Interactive comments on "Adsorption and Desorption studies of Delonix regia pods and leaves: Removal and recovery of Ni(II) and Cu(II) ions from aqueous solution" by Bolanle M. Babalola et al.

The author's responses to the Interactive comments (RC1) on Bolanle M. Babalola et al. are as follows:

Anonymous Referee #1 General comments: Comments from referee A clear objective

(and knowledge gap) at the end of the introduction is still missing or weak. How does this relate to previous research in the area? Author's response A clear objective of the study has been included at the end of the introduction and the way it relates to previous research have been included (now in line 106-118). Comments from referee The"therefore" in line 99 is not well underpinned/ Author's response The "therefore" is now well underpinned (now in line 118). Comments from referee Language, including tenses, is of poor quality. - Redundant information should be deleted. Author's response Tenses have been revised and redundant information have been deleted. Comments from referee The discussion of the results is weak. How do the results link to other research? How are the results (including isotherm constants and kinetic constants) relate to other adsorbents? Author's response The results on the isotherm and kinetic studies have been linked and supported with other reported work (now in line 306-340; line 350-418). The kinetics, isotherm and removal efficiency of D. regia biomass are now compared with other agro-waste (e.g. now in line 337-340, Table 3 has been created for comparison, etc). Comments from referee The order of the presented results should be reconsidered. Better to start with the kinetic tests and equilibrium tests and then relate that to change in pH and structure of the pods and leaves. Author's response The order in which the experiment was conducted was reported. i.e. parameters where optimised at different stages, therefore, the rearrangement of the sequence will distort the manuscript.

Specific comments: Comments from referee Line 15: include stating that it were "batch tests" Author's response Batch test has been mentioned in line 14 Comments from referee Line 17-18: it is not relevant in the abstract if data fit a model. More important is how well the metals adsorb in relation to other adsorbents. Author's response How well Cu (II) and Ni (II) adsorb on the adsorbents has been mentioned in the abstract (line 22-26). Moreso, the authors have also added if a data fit a model, as this is the norm in all adsorption papers. Comments from referee Line 21-22: delete sentence, since a good economic study is not made Author's response The sentence (line 21-22) supports our studies and we have not deleted it. However, we have avoided the use of

the word "economics" in the entire manuscript.

Grammatical errors where checked and corrected in the entire manuscript. i.e Line 20: "concentrations" was been corrected Line 38: "of heavy metals" have been deleted Line 39: "and cause serious pollution" deleted Line 43: supply revised to resources Line 44: "production" ( of glass, textiles, paper etc.) have been added Line 45: "apart from.. ecosystem" deleted Line 46: "human" (gills, liver: : :.) cannot be inserted, as the sentence refers to marine animals and not human Line 48: during = in the; of animal = in animals; have been revised Line 52: "also" (reported) has been added Line 53: "Like most heavy metals" deleted Line 54: "," after "health" has been added Line 55: "," after "compounds" and after "nickel", have been added Line 58-61 have been revised Line 64: "," after "value" has been added Line 66 and 67: "shells" has been corrected Line 68: "treat" = "remove" have been corrected Line 71: "," after "biomass" inserted; "optimum conditions" = "dosage" have been corrected "at" was inserted before pH and the stirring speed was deleted Line 75-78: the sentence has been deleted Line 83: "seeds" has been revised Line 84: "the treatment : : :. To" has been revised; "reduce" = "reducing" has been revised Line 88-90: the sentence has been deleted Line 96-98: the sentence has been deleted

Comments from referee Line 99-: give knowledge gap, based on the previous research and explain reason to study selected material better. Author's response The knowledge gap based on previous research have been provided i.e. line 106-118.

Line 106-108: sentence has been deleted

Comments from referee Lin 111: give concentrations of stock solutions Author's response The concentrations of stock solutions was mentioned in line 116

Comments from referee Line 118-121: give pHs to be studied: give concentrations that are studied Author's response pHs to be studied, concentrations that are studied were now given in line 153-162

Line 129: "use" = "used" has been revised Line 130-131: "is" = "was" (three times) have been revised Line 132: "are" = "were" has been revised

Comments from referee Line 129-137: explain procedure of the batch tests in detail (number of jars, concentrations, dosages); further explain the modelling part in the M&M section too. Author's response The procedure of the batch tests has been reported in detail. The modelling part now transferred to the M&M section

Line 141: "structures, being potentially beneficial for the uptake" has been revised Line 143-144: "is" ="was" (two times) have been corrected

Comments from referee Line 144-145: what do these data indicate? Give discussion with literature Author's response The EDS data represent the composition of the sample. However, It is used for qualitative analysis and not quantitative analysis, therefore there is no need to give a detail discussion of EDS. If it were to be XRF which provides the true quantitative analysis of the samples, a detail discussion might be required.

Comments from referee Line 156: how can we find the value in the Figure? - Line 158-159: why the authors have this opinion? Can they support this with literature? Author's response The XRD has been replotted and discussed anew. The different peaks and what they correspond to, has been indicated in the new plot.

Line 167: "reveals" has been revised; "significant" has been deleted Line 168: "adsorbents" has been revised ; "very" (subjective) has been deleted; "increased" has been revised Line 169: sentence has been deleted Line 170: "is" = "was' has been revised; "as the optimum pH and used" has been deleted Line 173: ", thus being in line with our findings" has been added

Comments from referee Line 175 (and further): do not connect data points with lines in figures (no meaning) Author's response The lines connecting data have been removed in figures, except the trendline in the isotherm and kinetic plots.

Line 180: "for Cu(II) ions" has been deleted; "pH 4 was used : : : pods while pH 5 was"

= "which were further" has been revised. Line 181: "leaves and pods, respectively" has been revised

Comments from referee Line 182-184: discuss with literature what happened. Author's response The results on the effect of pH has been discussed with literature

Line 184: "Thus: : : studies" has been deleted; "literatures" = "authors" has been revised Line 187: "Generally: : :. Media" has been deleted Line 188: reference has been given Line 189-191: ""when: : :.. particles" has been deleted Line 192: "the positive metal ions" has been added; "decreases" has been revised Line 194: "This will: : : adsorbent" has been deleted Line 201: "for" = "of" has been revised Line 203: the lines connecting data in Fig 5a have been removed, except the trendlines for the kinetic plot (5b).... the trendline is important for the displayed equations and R2 values. Also the lines connecting data in Fig 6 a, 7 and 8 were also removed Line 209: "interacting" = "interaction with" has been revised Line 212: "adsorbents" has been revised Line 213: "to" = "until" has been revised Line 217: How the work of Hansen supports the data has been briefly explained Line 221: "is a: : : fact" = "indicates" has been revised; reference for this statement has been provided.

Comments from referee Line 223-226: should be in Materials and methods section Author's response The kinetic and isotherm equations have been transferred to the Materials and methods section

Line 225: "any" has been deleted Line 235: "decreased" has been revised; references have been given Line 246: "higher number" = "high concentration" has been revised; reference for this statement has been given Line 248-249: the sentence has been deleted

Comments from referee Line 250:263: should be explained in Materials and methods section Author's response Line 250:263: now explained in Materials and methods section

Line 264- 267 has been revised Line 269: "The fit to the plot of the Langmuir isotherm suggests the possible monolayer: : : ha been revised Line 271: "slight" has been deleted Line 272-274: sentence has been delete Line 283: "The Freundlich isotherm is used to describe adsorptions onto.." has been revised Line 288-293 has been moved to Materials and methods section Line 321: "shows" has been revised Line 322: "biomasses" has been revised Line 322-325: the statement has been referenced Line 335: "concentrations" has been revised Line 336: "Was" = "were" has been revised Line 348: "application" = "use" has been revised Line 352-354 have been explained in the body of the manuscript

Comments from referee Line 357-359: this cannot be concluded because a thorough economic analysis is not made Author's response We have avoided the use of the word "economics". However, the fact stil remains that the biomass is an agricultural waste, the pods and leaves of D. regia litters the ground and rot. This has been mentioned in the introduction and concluding section.

The authors appreciate the editors and reviewers of this manuscript, the comments and suggestions received have greatly improve the quality of the manuscript.

Thank you.

---

## Author Comment (AC2) · 2 Apr 2020

Response to Interactive comments on "Adsorption and Desorption studies of Delonix regia pods and leaves: Removal and recovery of Ni(II) and Cu(II) ions from aqueous solution" by Bolanle M. Babalola et al.

The authors' responses to the Interactive comments (RC2) on Bolanle M. Babalola et al. are as follows:

Anonymous Referee #2 General comments: Comments from referee The approach is

interesting; however, the document has to be improved in the way of presenting the information/results, comparing with other literature and avoiding presenting the same data in both graph and tables. The graphs quality has to be improve. Author's response The manuscript has been improved in the way of comparing the results with other literature studies. The quality of the graphs has been improved.

Comments from referee It is recommended to compare with other adsorbents in terms of adsorption capacity (mg/g) and not percentage and including the initial concentrations. Author's response We have now compared the D. regia adsorption capacity with other agro-waste reported in the literature. A new table has been created in this regard (Table 3). However, we have chosen to use percentage as a measure of the efficiency based on the suggestion by reviewer #1.

Comments from referee The aim or need of the study is not well stated, the application is only for wastewater? Author's response The aim of the study is now well stated at the end of the introduction section.

Comments from referee The conclusions both, pods and leaves, are promising ones to remove the metals or only one of the plant parts? Author's response Both the pods and leaves could be used for the removal of Ni and Cu ions. However, the pods performed better in term of the amount of metal ions removed and regeneration of the biosorbent. These have been stated clearly in the manuscript as well as in the concluding section.

Comments from referee How should the material be applied, in batch or filters? Author's response We have conducted a batch experiment and our finding have showed that the agro-wastes are effective. We encourage other researchers to build on our studies by conducting the column/filter experiment. Comments from referee No study about economics was performed, so better not to mentions economic aspects. Author's response Since economic study was not performed, we have avoided the use of the term economics in the entire manuscript.

Specific comments: Comments from referee Line 13-14 mention that conditions such

as pH, contact time, metal ions concentration and the presence of sodium ions were studied, however in the abstract only the results of kinetics is presented, the results of the others conditions should be included. Author's response Abstract: the results of the other conditions have been included (now in line 17-19).

Comments from referee Line18-19: present the results of the isotherm data, at least the capacity in terms of mgMetal/gadsorbent and state is it is good or not in relation with others adsorbents in literature. Author's response The adsorption isotherm i.e the adsorption capacity of the pods and leaves of D. regia in terms of mgMetal/g adsorbent has been stated in the abstract.

Introduction: Comments from referee Include newer references as the newest one is from 2016. Author's response Newer references have been included

Comments from referee Some information about reports, monitoring or regulations of the metals in drinking and wastewater is needed, so, there will be an idea of which concentrations are of concern and should be used in the study to evaluate the adsorbents, that would justify why 100mgM/L was used in the study. Author's response The permissible limit of Cu(II) and Ni(II) in drinking water and wastewater has been included the manuscript (line 52 and 53; line 62 and 63).

Comments from referee Line 66-98 the removal capacity effectiveness of many adsorbents is reported, however almost no data of the capacity and initial concentrations of the metals is included, include the mg/g if that information is reported. That information gives an idea of why to go for natural adsorbents. Author's response Data on the capacity and initial concentration of metals have been included, the reported adsorption capacity reported in other studies in mg/g have been included.

Comments from referee Line 99 explain why is this research needed and why to choose this adsorbent, why to check the leaves and the pods? Author's response The reason why this research is needed and why we have chosen to use the pods and leaves of D. regia have now been detailed in the manuscript (line 106-118) Materials and methods:

Comments from referee Lines 111-112 were the leaves and pods dried? Author's response The leaves and pods were dried, this has been included in the manuscript (line 113)

Comments from referee Line 117 what were the concentrations used? Author's response The concentrations of metal ions used are now included (line 157).

Comments from referee Line 120-121 include the sodium and nitic acid concentrations tested. Author's response The sodium and nitic acid concentrations used are now mentioned in the manuscript (line 160).

Comments from referee Line 119 instead of referencing to paper include the procedures in detail including for example: type of water used, adsorbent dosage, metal concentrations, rpm, volume of solutions, time, etc. Besides it would be better to include subitems for each experiment starting from: kinetics, pH effect, isotherms, ionic strength effect and desorption experiment. Each one explained in detail. Author's response The procedures used in the adsorption experiment have been detailed in the manuscript (line 153 – 166).

Comments from referee Were the experiments performed at least in duplicate? Author's response The experiments were performed in triplicate, this was mentioned in the manuscript

Comments from referee Line 134 Equation 1 is not relevant, it is better to include after each subitem mentioned in the previous comment the corresponding equations for kinetics and isotherms. Better here than in the results and discussion section. Author's response All equations regarding kinetics and isotherm are now transferred to the materials and methods section. Moreover, Equation 1 is very important, and we have not deleted it.

Comments from referee Line 136 a specific subsection is needed for the analitycal procedure including equipment brand and model and the detection limit for both metals.

Author's response A new subsection on analytical procedure has been included (line 168-179).

Results and discussion: Comments from referee Lines 156-158 The diffractogram first shows that both materials are amorphous, that could be related to high surface area and potential adsorption. Besides, include in Figure 3 the diffractogram of cellulose so, that it can be directed compare, specially thinking that in natural products not only cellulose is present, what about lignin for example? DRX is not enough to characterize a natural products, other techniques like IR, MNR, etc is needed. What is found in literature for natural adsorbents of this type? Which moiety are responsible for the adsorption? Line 157 mention amino group, but in figure 2 no nitrogen was detected. Author's response The diffractogram have been replotted and all phases present have been identified based on the ICSD Nos. 03-0289, 20-0231 and 26-1077 and discussed.

Comments from referee Line 164-165 start with the kinetic study. Author's response We have chosen to report the research in the order in which the experiments were conducted i.e. pH was optimized, followed by contact time, etc.

Comments from referee Line 165 it is important to mention that in general pods performed better than leaves for both metals in the whole pH range, why? Author's response We have now mentioned that the pods performed better than leaves for both metals (line 416 -418), and also in the concluding section.

Comments from referee Line 176: include adsorbent dose. Author's response Adsorbent dosage has been included in the experimental conditions mentioned in all of the adsorption plots.

Comments from referee Line 183: what can be the precipitate? What literature says? Author's response The nature of the precipitates has been included (line 277).

Comments from referee Line 200 the equipment and time evaluated goes in the materials and methods section. Line 201 it is evident that pods are better than leaves, make

a comment. Author's response The equipment and time evaluated have been moved to the materials and methods section. Comments have been made on the fact that the pods are better than leaves (line 309 -310).

Comments from referee Line 207 include adsorbent dose. Author's response Adsorption dosage has been included in the experimental conditions mentioned in the plot

Comments from referee Line 205 -228 the information in Fig. 5b and Table 1 is basically the same, let the Table 1 in the document. Author's response The authors have chosen to retain the kinetic and isotherm plots, unless the editor insist that the plots should be removed.

Comments from referee Line 210-213 then adsorption was mainly in 30 min, how is that compare to the literature? Faster, slower? Author's response The exceptionality of 30 min contact time for D. regia biomass in comparison to other agro-waste has been detailed in the manuscript (line 312-320).

Comments from referee Line 218 how Hansen et al. (2010) supports your results on kinetics? Author's response How Hansen et al. (2010) supports the results on kinetics have been included.

Comments from referee Line 223: put the equation in materials an methods Author's response All equations have been transferred to the materials and methods.

Comments from referee Line 228: compares the Qe and k values obtained for the pods and the leaves among them and the literature. Author's response The Qe and k values obtained for the pods and the leaves in the literature has been included (line 337-340).

Comments from referee Line 230 this is confusing is that the same experiment for the isotherms? Then, the title item should refers to the isotherms experiments. Author's response The data obtained on the effect of initial adsorbate concentration was used to model the isotherm. The title now reflects isotherms experiments (line 342).

Comments from referee Line 241 regarding with Fig. 6b and Fig 6c those are the

linearized isotherms and the same information is on Table 2. Recommend to plot Qe vs Ce instead of those figures. Author's response The authors have chosen to retain the kinetic and isotherm plots, unless the editor insist that the plots should be removed.

Repeated word "correlation" has been deleted.

Comments from referee Line 296-300 the data suggest that the pods are better than the leaves, it can be seen from the graphs in Figs. 5a and 6a, from the data. Besides, 1/n and Kf are lower and higher respectively for the pods for both metals confirming that the pods perform better and adsorption is more favorable on them at the concentrations studied. Author's response The fact that 1/n and Kf are lower and higher respectively for the pods for both metals confirming that the pods perform better, have been included in the manuscript (line 416 -418). Also reinstated in the manuscript is that Figs. 5a and 6a showed that the pods is better than the leaves.

Comments from referee Line 300-304 higher R2 only confirms which data fit better to the model not which one is a better or a favorable adsorption. Author's response The wrong notion of a higher R2 associated to a favorable adsorption has been corrected.

Comments from referee Line 305 the adsorbent dose experiment is to determine the isotherm. Author's response The adsorbent dose experiment can not be used to determine the isotherm, but the initial adsorbate concentration experiment.

Comments from referee Line 316 better to use normal scale, not logarithmic so the effect is easier to see. Author's response Normal scale has been used to plot Fig 7b.

Comments from referee Line 331 was that experiment as all made by duplicate? Author's response All experiments were conducted in triplicate.

Comments from referee The results with 0.05M HNO3? Are confusing, how is that the Ni and Cu pods % is substantially lower and with ultra pure water and higher concentration is practically the same in both cases? How to explain that? Author's response Section 3.3 has been re-discussed.

Comments from referee Line 352-352 what can be concluded from the isotherm data? Are those good or not for the removal of the metals, are them better than others? Line 354/355 what to conclude from the kinetic, is it fast or lower than others? Author's response The isotherm discussion is detailed in section 3.2.3., a brief statement of the isotherm study is required in the concluding section and this is what has been done.

Comments from referee Line 355/356 see comment on line 331, confusing data. Author's response The desorption study has been revisited and re-discussed.

Comments from referee Line 357-359 no economical study was done, so can't conclude that. Besides only two metals were evaluated and using ultrapure water, so can't conclude is application in natural conditions and besides, can't conclude about others environmental contaminants Author's response We have avoided the use of the term "economics"

The authors appreciate the editors and reviewers of this manuscript, the comments and suggestions received have greatly improve the quality of the manuscript.

Thank you.

---

## Author Response (AR1)

Dear Editor,

The response to the reviewer's comments are as follows:

**Reviewer #1**

General comments:
- line 104-118

A clear objective of the study has been included at the end of the introduction and the way it relates to previous research have been included.

The "therefore" now in line 118 is now well underpinned.

- Tenses have been revised and redundant information have been deleted.

- line 306-340; line 350-418

The results (i.e isotherm and kinetics) have been linked and supported with other research

- line 337-340, Table 3, etc

The kinetics, isotherm and removal efficiency of *D. regia* biomass is now compared with other agro-waste

- The order in which the experiment was conducted was reported. i.e. parameters where optimised at different stages, therefore, rearranging the sequence will distort the manuscript.

Specific comments:

- Batch test has been mentioned in line 14

- pg 1, line 22-26
how well Cu (II) and Ni (II) adsorb on the adsorbents has been mentioned in the abstract.

- pg 1, line 21 -22
The sentence (line 21-22) supports our studies and we have not deleted it. Moreso, the biomass is an agricultural waste and cheap. However, we have avoided the use of the word "economics".

- Grammatical errors where checked and corrected in the entire manuscript.
i.e Line 20: "concentrations" was been corrected
Line 38: "of heavy metals" have been deleted
Line 39: "and cause serious pollution" deleted
Line 43: supply revised to resources
Line 44: "production" ( of glass, textiles, paper etc.) have been added
Line 45: "apart from.. ecosystem" deleted
Line 46: "human" (gills, liver: : :.) can not be inserted, as the sentence refers to marine animals and not human
Line 48: during = in the; of animal = in animals; have been revised
Line 52: "also" (reported) has been added

Line 53: "Like most heavy metals" deleted
Line 54: "," after "health" is added
Line 55: "," after "compounds" and after "nickel", added
Line 58-61 have been revised
Line 64: "," after "value" has been added
Line 66 and 67: "shells" corrected
Line 68: "treat" = "remove" have been corrected
Line 71: "," after "biomass" inserted; "optimum conditions" = "dosage" have been corrected
"at" was inserted before pH and the stirring speed deleted
Line 75-78: sentence deleted
Line 83: "seeds" resvised
Line 84: "the treatment : : :. To" revised; "reduce" = "reducing" revised
Line 88-90: sentence deleted
Line 96-98: sentence deleted
Line 99-:
The knowledge gap based on previous research have been provided i.e. line 106-118.
Line 106-108: sentence deleted
Lin 111: the concentrations of stock solutions was mentioned in line 116
Line 118-121: pHs to be studied, concentrations that are studied were now given in line 153-162
Line 129: "use" = "used" revised
Line 130-131: "is" = "was" (three times) revised
Line 132: "are" = "were" revised
Line 129-137: the procedure of the batch tests has been reported in detail.
The modelling part now explained in the M&M section
Line 141: "structures, being potentially beneficial for the uptake" revised
Line 143-144: "is" ="was" (two times) corrected
Line 144-145 the EDS data represent the composition of the sample. However, It is used for qualitative analysis and not quantitative analysis, therefore there is no need to give a detail discussion of EDS. If it were to be XRF which provides the true quantitative analysis of the samples, a detail discussion might be required.
Line 156, Line 158-159: The XRD has been replotted and discussed anew. The different peaks and what they correspond to, has been indicated in the new plot
Line 167: "reveals" has been revised; "significant" deleted
Line 168: "adsorbents" revised ; "very" (subjective) was deleted; "increased" revised
Line 169: sentence deleted
Line 170: "is" = "was' revised; "as the optimum pH and used" deleted
Line 173: ", thus being in line with our findings" added
Line 175 lines connecting data have been removed in figures, except the trendline in the isotherm and kinetic plots.
Line 180: "for Cu(II) ions" has been deleted; "pH 4 was used : : : pods while pH 5 was" = "which were further" has been revised.
Line 181: "leaves and pods, respectively" revised
Line 182-184: the results on the effect of pH has been discussed with literature
Line 184: "Thus: : : studies" deleted; "literatures" = "authors" revised
Line 187: "Generally: : :. Media" deleted
Line 188: reference given
Line 189-191: ""when: : :.. particles" deleted
Line 192: "the positive metal ions" added; "decreases" revised
Line 194: "This will: : : adsorbent" deleted

Line 201: "for" = "of" revised

Line 203:

the lines connecting data in Fig 5a have been removed, except the trendlines for the kinetic plot (5b).... the trendline is important for the equations and $R^2$ values displayed.

Also the lines connecting data in Fig 6 a, 7 and 8 were also removed

Line 209: "interacting" = "interaction with" revised

Line 212: "adsorbents" revised

Line 213: "to" = "until" revised

Line 217: how the work of Hansen supports the data has been explained

Line 221: "is a: : : fact" = "indicates" revised; reference for this statement has been given

Line 223-226: transferred in Materials and methods section

Line 225: "any" deleted

Line 235: "decreased" revised; references are given

Line 246: "higher number" = "high concentration" revised; reference for this statement was given

Line 248-249: sentence deleted

Line 250:263: now explained in Materials and methods section

Line 264- 267 revised

Line 269: "The fit to the plot of the Langmuir isotherm suggests the possible monolayer: : : revised

Line 271: "slight" deleted

Line 272-274: delete sentence

Line 283: "The Freundlich isotherm is used to describe adsorptions onto.."

Line 288-293 moved to Materials and methods section

Line 321: "shows" revised

Line 322: "biomasses" revised

Line 322-325: the statement now referenced

Line 335: "concentrations" revised

Line 336: "Was" = "were" revised

Line 348: "application" = "use" revised

Line 352-354 have been explained in the body of the manuscript

Line 357-359: We have avoided the use of the word "economics", however, the fact remains that the biomass is an agricultural waste, the pods and leaves of *D. regia* litters the ground and rot. This has been mentioned in the introduction and concluding section.

**Reviewer #2**

General comments

The manuscript has been improved in the way of comparing the results with other literature studies.

The quality of the graphs has been improved.

We have now compared the D. regia adsorption capacity with other agro-waste reported in the literature. A new table has been created in this regard (Table 3). However, we have chosen to use percentage as a measure of the capacity based on the suggestion by reviewer #1.

The aim of the study is now well stated at the end of the introduction section.

The conclusions, both the pods and leaves could be used for the removal of Ni and Cu ions. However, the pods performed better in term of the percentage metal ions removed and regeneration of the biosorbent. These have been stated clearly in the concluding section.

We have conducted a batch experiment and our finding have showed that the agro-wastes are effective. We encourage other researchers to build on our studies by conducting the column/filter experiment.
Since economic study was not performed, we have avoided the use of the term economics.

Specific comments
Abstract: the results of the other conditions have been included (line 17-19). The adsorption isotherm i.e the adsorption capacity of the pods and leaves of D. regia in terms of mgMetal/g adsorbent has been stated in the abstract.

Introduction:
Newer references have been included
The permissible limit of Cu(II) and Ni(II) in drinking water and wastewater has been included the manuscript (line 52 and 53; line 62 and 63).
The reason why this research is needed and why we have chosen to use the pods and leaves of D. regia have now been detailed in the manuscript (line 106-118)

Materials and methods:
The leaves and pods were dried, this has been included in the manuscript (line 113)
The concentrations of metal ions used are now included (line 157).
The sodium and nitic acid concentrations used now mentioned in the manuscript (line 160).
The procedures used in the adsorption experiment have been detailed in the manuscript (line 153 – 166)
The experiments were performed in triplicate, this was mentioned in the manuscript
All equations regarding kinetics and isotherm are now transferred to the materials and methods.
A subsection on analytical procedure has been included (line 168-179).

Results and discussion:
The diffractogram have been replotted and all phases present have been identified.
We have chosen to report the experiment in the order in which the experiment was conducted i.e. pH was optimized, followed by contact time, etc.
We have now mentioned that the pods performed better than leaves for both metals (line 416 - 418), also in the concluding section.
Adsorbent dosage has been included in the experimental conditions mentioned in all of the adsorption plots.
The nature of the precipitates has been included (line 277).
The equipment and time evaluated have been moved to the materials and methods section.
Comments have been made on the fact that the pods are better than leaves (line 309 -310).
Adsorption dosage included in the experimental conditions mentioned in the plot
The uniqueness of 30 min contact time for D. regia biomass in comparison to other agro-waste has been detailed in the manuscript (line 312-320)
How Hansen et al. (2010) supports the results on kinetics have been included.
All equations have been transferred to the materials and methods.
The Qe and k values obtained for the pods and the leaves in the literature has been included (line 337-340).
The data obtained on the effect of initial metal ions concentration is often used to model the isotherm. The title now reflects isotherms experiments (line 342).
The authors have chosen to retain the kinetic and isotherm plots, unless the editor insist that the authors remove these plots.
Repeated word "correlation" has been deleted.

The fact that 1/n and Kf are lower and higher respectively for the pods for both metals confirming that the pods perform better have been included in the manuscript (line 416 -418). The wrong notion of a higher R2 and a favorable adsorption has been corrected.

The adsorbent dose experiment can not be used to determine the isotherm, but the initial adsorbate concentration experiment.

Normal scale is now used to plot Fig 7b.

All experiments were conducted in triplicate.

Section 3.3 has been re-discussed.

The isotherm has been detailed in section 3.2.3, a brief of the isotherm study is required in the concluding section and this is what has been done.

The desorption study has been revisited and re-discussed.

We have avoided the use of the term "economics"

An author (Olayide S Lawal) was initially omitted and has been added, moreover, Olushola S Ayanda is now the last author on the list.

We appreciate the editors and reviewers of this manuscript, the comments and suggestions received have greatly improve the quality of the manuscript.

Thank you.

Dr Olushola S Ayanda.

[revised manuscript text omitted]

---

## Author Response (AR2)

Dear Editor,

The response to the Topical Editor's comments are as follows:

The misspelled Lagmuir has been corrected

Italicized texts have been crosschecked, the italicized texts in the manuscript are scientific names e.g. *Delonix regia* which must be written in italics.

How the Freundlich and Langmuir isotherms relate to other agrowaste has been indicated on page 16 of the manuscript.

The dosages in relation to the removal percentages has been included in Table 3 (page 18). We have also included the concentration of the adsorbates that were used.

We appreciate the Topical editor of this manuscript, the comments and suggestions raised have greatly improve the quality of the manuscript.

Thank you.

Dr Olushola S Ayanda.

[revised manuscript text omitted]